# Shift current photovoltaic effect in a ferroelectric charge-transfer complex

M. Nakamura [1,2], S. Horiuchi[3], F. Kagawa [1], N. Ogawa[1], T. Kurumaji[1], Y. Tokura[1,4] & M. Kawasaki[1,4]

Shift current is a steady-state photocurrent generated in non-centrosymmetric single crystals and has been considered to be one of the major origins of the bulk photovoltaic effect. The mechanism of this effect is the transfer of photogenerated charges by the shift of the wave functions, and its amplitude is closely related to the polarization of the electronic origin. Here, we report the photovoltaic effect in an organic molecular crystal tetrathiafulvalene-$p$-chloranil with a large ferroelectric polarization mostly induced by the intermolecular charge transfer. We observe a fairly large zero-bias photocurrent with visible-light irradiation and switching of the current direction by the reversal of the polarization. Furthermore, we reveal that the travel distance of photocarriers exceeds 200 μm. These results unveil distinct features of the shift current and the potential application of ferroelectric organic molecular compounds for novel optoelectric devices.

[1] RIKEN Center for Emergent Matter Science (CEMS), Wako 351-0198, Japan. [2] PRESTO, Japan Science and Technology Agency (JST), Kawaguchi 332-0012, Japan. [3] Flexible Electronics Research Center (FLEC), National Institute of Advanced Industrial Science and Technology (AIST), Tsukuba 305-8565, Japan. [4] Department of Applied Physics and Quantum Phase Electronics Center (QPEC), University of Tokyo, Tokyo 113-8656, Japan. Correspondence and requests for materials should be addressed to M.N. (email: masao.nakamura@riken.jp)

Photocurrent generation is the most essential process for solar cell action and consists of two steps: the creation of electron–hole pairs by light irradiation and the spatial separation of electrons and holes. In traditional solar cells, the latter process is driven by the built-in electric field at the interface of p–n junctions. Efficient carrier extraction requires high carrier mobility and long carrier lifetime in the constituent materials as well as careful control of the interface, which make the development of novel photovoltaic materials challenging.

An alternative approach is to employ materials that lack inversion symmetry. The generation of steady-state photocurrent in non-centrosymmetric crystals is known as the bulk photovoltaic effect (BPVE)[1, 2]. This phenomenon was discovered more than half a century ago, and the origin was initially presumed to be the internal electric field associated with the bulk polarization[3]. However, soon later it was revealed that the direction of the photocurrent depends on the light polarization and photon energy, as observed in ferroelectric $BaTiO_3$[4]. Furthermore, the polarization was observed to be not a necessary condition for BPVE[1]. To provide consistent explanations for these features of BPVE, a theory called the shift current model was proposed, which attributes the charge separation to the quantum mechanical process arising from the asymmetry in the electronic wave functions instead of the internal field[5, 6]. Clear evidence for this model was provided by the demonstration that the theoretical calculation of the shift current can quantitatively reproduce the complex photocurrent spectra of $BaTiO_3$[7]. Recently, there have been renewed interests in BPVE and, in particular, of the shift current because of the significant advances in the theoretical understanding on its topological aspects[8–11] and in the synthesis and computational design of new materials exhibiting large shift current for visible light[12–18].

Shift current generation is a second-order non-linear optical process[5, 6]. The form of the response function implies that the position of the electron wave packet immediately shifts in real space upon the interband optical transition. The average distance and direction of the shift, called the shift vector, is given by the difference in the Berry connection of the Bloch wave functions of the two corresponding bands and has a non-zero value only when the inversion symmetry is broken. Although a larger polarization does not necessarily lead to a larger shift current, the polarization and shift current are closely related. The spontaneous polarization generally has two contributions: the ionic component ($P_{ion}$), which originates from the displacement of charged atoms or molecules, and the electronic one ($P_{el}$) arising from the asymmetry in the wave function forming the covalent bonds. In the modern theory of polarization, $P_{el}$ can be quantified using the Berry phase for the valence states[19, 20]. The shift vector is also expressed using the geometric phase, and its form indicates the difference in $P_{el}$ between the conduction and valence band, i.e., the change in $P_{el}$ induced by the optical transition[11]. Therefore, a material hosting a large $P_{el}$ is a promising candidate showing a large shift current.

Here, we report the photovoltaic properties of an organic charge-transfer (CT) complex, tetrathiafulvalene-p-chloranil (TTF-CA), from the viewpoint of the shift current. TTF-CA consists of a face-to-face stack of alternating electron donor (D = TTF) and electron acceptor (A = CA) molecules along the a-axis (Fig. 1a)[21–23]. Because this compound has a valence instability arising from the subtle balance between the ionization energy and electrostatic Madelung energy, it undergoes a transition from a neutral (N = $D^0A^0$) to an ionic (I = $D^+A^-$) phase at the Curie temperature $T_C = 81$ K, as depicted in Fig. 1b. In actuality, the ionicity ($\equiv \rho$) in the $D^{+\rho}A^{-\rho}$ chain is ~0.3 in the N phase and ~0.7 in the I phase because of the CT interaction between D–A molecules[24–26]. Concomitantly with the N–I transition, the quasi-

one-dimensional (1D) nature along the stacking axis yields Peierls-type or spin-Peierls-type pairwise lattice deformation[27, 28]. This dimerization inevitably leads to the polar state, and the polarization reversal, indicating the proper ferroelectricity, has recently been experimentally verified[29]. First-principles calculations and detailed structural analyses have revealed that most of the polarization arises from the CT between neighboring D–A molecules and that $P_{el}$ is more than 20 times larger than $P_{ion}$; hence, it is regarded as one of the representative electronic ferroelectrics[29–31]. Thus, we expect an enhancement of the shift current in TTF-CA because of the large $P_{el}$. Furthermore, the basic physical properties of TTF-CA can be represented by the Rice–Mele model, which is a 1D tight-binding chain that includes the staggered onsite potentials and the inter-and intra-dimer charge transfers[32]. Based on this model, the polarization, shift current, and their relation have been intensively investigated[8, 9, 11, 19, 33, 34]. Therefore, TTF-CA can provide an ideal platform with simple but essential conditions for the shift current generation. We examine the photovoltaic properties of TTF-CA and observed that sizable zero-bias photocurrent appears with visible-light irradiation in the I phase. We also reveal an anomalously long travel distance of photocarriers. These results provide clear indications of the shift current first observed in organic CT complexes.

## Results

**Temperature dependence of polarization in TTF-CA.** We first examined the polarization property in a single crystal of TTF-CA. Figure 1c shows the temperature dependence of the polarization deduced from the pyroelectric current measured after the crystal was poled in an electric field of 2 kV cm$^{-1}$. The spontaneous polarization abruptly appears at $T_C$ and increases above

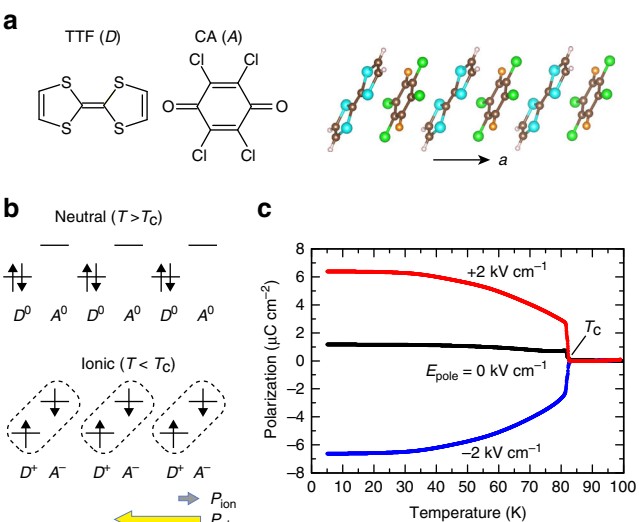

**Fig. 1** Molecular structures and neutral-to-ionic phase transition in TTF-CA. **a** Molecular structure of tetrathiafulvalne (TTF) and p-chloranil (CA) and the mixed stack structure of the two molecules along the a-axis. TTF is an electron donor (D) and CA is an electron acceptor (A). **b** Schematic electronic structures of TTF-CA in neutral and ionic phases. In the ionic phase, the dimer formation induces spontaneous polarization. The polarization has ionic and electronic contributions ($P = P_{ion} + P_{el}$). The former originates from the displacement of charged molecules, while the latter originates from the charge transfer between D–A molecules. In TTF-CA, $P_{el}$ is much larger than $P_{ion}$ and their directions are opposite. **c** Temperature dependence of the spontaneous polarization determined from the pyroelectric current measured after the sample was cooled under poling fields ($E_{pole}$) of ±2 kV cm$^{-1}$ and without the poling procedure ($E_{pole} = 0$)

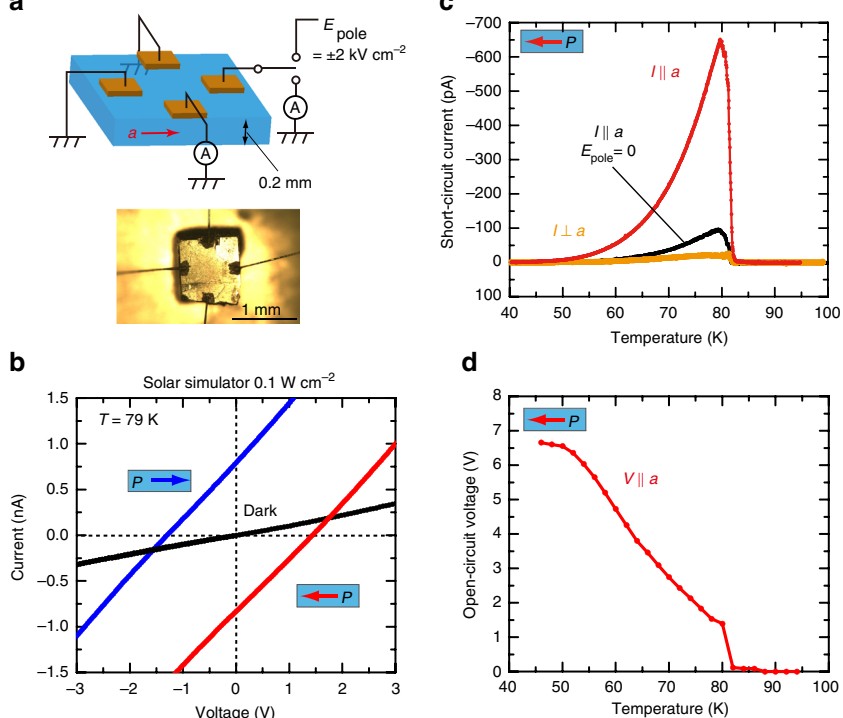

**Fig. 2** Photovoltaic properties of TTF-CA. **a** The *upper panel* presents a schematic of the sample configuration for the photovoltaic effect measurements. The *lower panel* presents an optical microscope image of an actual sample. **b** I–V characteristics at 79 K with and without photoirradiation. The light source was a solar simulator (0.1 W cm$^{-2}$). Before the I–V measurements, the polarization was aligned by cooling the sample from 100 to 30 K under $E_{pole}$ = 2 kV cm$^{-1}$. The signs of the photocurrent and photovoltage are reversed by inverting $E_{pole}$. **c** Temperature dependence of zero-bias photocurrent measured under photoirradiation. The measurements were performed along the directions parallel and perpendicular to the *a*-axis after the same poling procedure. We also measured the photocurrent without the poling procedure ($E_{pole}$ = 0). **d** Temperature dependence of the open-circuit photovoltage along the *a*-axis

6 μC cm$^{-2}$ at the lowest temperature. This polarization value is consistent with the reported value derived from P–E curves[29]. The polarization direction can be reversed by applying the opposite poling field. These results indicate that a single-domain ferroelectric state can be obtained by the poling field.

**Photovoltaic properties of TTF-CA.** Single crystals of TTF-CA with dimensions of approximately 1 × 1 × 0.2 mm were used to characterize the photovoltaic properties. Two pairs of electrodes were formed orthogonally with each other on the single crystal with carbon paste, as illustrated in Fig. 2a, to measure the photocurrent parallel and perpendicular to the *a*-axis. White light from a solar simulator with an intensity of 0.1 W cm$^{-2}$ was used for the excitation. Before the measurements of the photovoltaic properties, the single polar domain state was prepared by applying an electric field of 2 kV cm$^{-1}$ in the cooling process. Figure 2b presents the current (I)–voltage (V) characteristics along the *a*-axis measured at a temperature of 79 K just below $T_C$. The results indicate a finite zero-bias (short-circuit) photocurrent under photoirradiation (Fig. 2b). The sign of the photocurrent changes with the reversal of the polarization such that the photocurrent and polarization maintain an antiparallel relation. The obtained zero-bias photocurrent density was 1.6 μA cm$^{-2}$, which is much larger than those in single crystals of other visible-light-responsive ferroelectric compounds, such as BiFeO$_3$ (0.4 μA cm$^{-2}$ for laser light (hν = 3.06 eV, 40 W cm$^{-2}$)[35]), and a layered orga-nometallic perovskite halide (0.055 μA cm$^{-2}$ for monochromized light (hν = 2.95 eV, 0.08 W cm$^{-2}$)[36]) measured in the lateral configuration. We also verified that the magnitude of the dc photocurrent was constant for at least several hours at a constant temperature, indicating that the photocurrent is a steady-state current.

The observed photovoltaic effect is closely related to the spontaneous polarization, as evident from its temperature dependence. The zero-bias photocurrent parallel to the *a*-axis is absent in the N phase, abruptly emerges when the I phase sets in at $T_C$, and gradually decreases upon further decreasing the temperature (Fig. 2c). The open-circuit photovoltage also appears from $T_C$ and monotonously increases toward lower temperatures as opposed to the photocurrent (Fig. 2d). The photovoltage reaches 6.5 V at 50 K, which is ~10 times larger than the CT energy gap of TTF-CA. The above-bandgap photovoltage is a clear signature of the bulk photovoltaic effect[37]. Another striking feature of the observed photocurrent is the large anisotropy; the photocurrent perpendicular to the *a*-axis is far smaller than that in the parallel direction, as shown in Fig. 2c. It is expected from symmetry considerations that unpolarized light can induce a zero-bias photocurrent only along the polar axis. The observed small photocurrent perpendicular to the *a*-axis is most likely caused by a slight misalignment of the electrodes.

**Optical conductivity and photocurrent action spectra.** To see the spectral distribution of the photocurrent, we performed spectroscopic measurements. Figure 3a presents the polarized optical conductivity spectra of TTF-CA in the I phase (at 79 K) obtained by the Kramers–Kronig analysis of the reflectivity spectra. The spectrum for the electric field vector parallel to the stack axis (E//a) exhibits a strong peak at $E_{CT}$ = 0.5 eV, which is attributed to the CT exciton formation[23, 24, 38]. Figure 3b presents the photocurrent action spectra measured with polarized lights. Reflecting the narrow CT gap of TTF-CA, the spectra spread across a wide photon energy range from the near-infrared to ultraviolet region. The zero-bias photocurrent appears from an energy slightly higher than $E_{CT}$, indicating that the dissociated

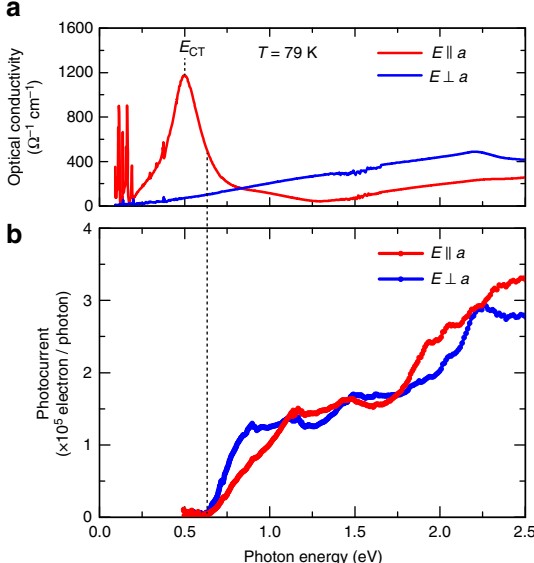

**Fig. 3** Optical conductivity and photocurrent action spectra. **a** Polarized optical conductivity spectra of TTF-CA measured in the I phase ($T = 79$ K). $E_{CT}$ denotes the energy of the CT excitons. The several peaks observed in the low photon energy region are related to the intramolecular vibration modes of TTF and CA molecules. **b** Photocurrent action spectra for light polarizations parallel and perpendicular to the $a$-axis obtained at the same temperature. The onset photon energy of the photocurrent is 0.6 eV, which is slightly higher than $E_{CT}$

electrons and holes are less generated by the resonant CT excitation[39]. The spectrum amplitude for $E//a$ light polarization gradually increases for higher photon energy excitation, perhaps because the dissociation of the CT excitons is accelerated.

**Position dependence of the photocurrent for local excitation.** In the shift current, charges are carried by the coherent evolution of the Bloch wave functions, which propagate rapidly, while being less affected by charge carrier recombination and phonon scattering processes[40]. This is in a clear contrast to those for $p$-$n$ junctions, in which carriers travel to the electrode via drift and diffusion transport and, hence, their mobilities are important for the carrier extraction. To highlight the distinct feature of the carrier transport in the shift current, we examined the position dependence of the zero-bias photocurrent by irradiating a focused laser light and scanning it between the electrodes, as depicted in the inset of Fig. 4b. The photon energy was 2.3 eV, and the spot size was $10 \times 200$ μm. The measurements were performed both in the I phase (at 70 K) and the N phase (at 90 K), as displayed in Fig. 4a and b, respectively. In the I phase, the photocurrent amplitude showed a rather flat profile as long as the laser spot was located on the sample. In contrast, the photocurrent in the N phase showed a maximum at the sample–electrode interfaces and opposite signs between the left-and right-hand interfaces.

The profile observed in the I phase clearly indicates that the photocurrent arises from the bulk effect and not from the interfacial effect. Several possible mechanisms have been proposed for the photocurrent generation in single-domain ferroelectric compounds other than the shift current: asymmetric band bending formed near the electrodes[41], a depolarization field caused by the imperfect screening of the surface charges[41, 42], and the existence of asymmetric scattering centers or electrostatic potentials[1, 2]. The result of the position dependence of the photocurrent in the I phase rules out these possibilities because the current arising from the first mechanism would be enhanced

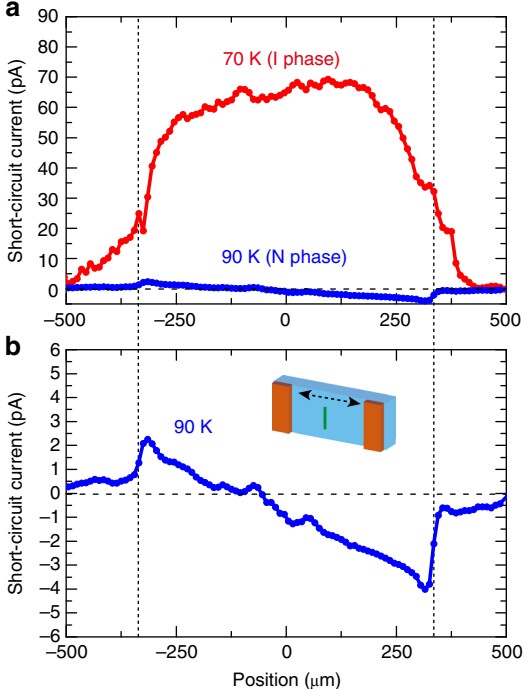

**Fig. 4** Position dependence of photocurrent induced by local excitation. **a** Position dependence of photocurrent measured in the I phase (70 K) and the N phase (90 K). The dotted lines denote the positions of the sample–electrode interfaces. **b** Position dependence in the N phase in Fig. 4a expanded along the vertical axis. The inset presents a schematic of the measurement set-up. A laser light with a wavelength of 532 nm was focused into a rectangular shape ($10 \times 200$ μm) and scanned across the sample. The interelectrode distance was ~670 μm

near the sample–electrode interfaces and both the second and third mechanisms would be effective only for short interelectrode spacing, typically below 100 nm[2, 41]. The extraction of carriers generated more than 200 μm away from the electrodes despite the highly correlated electron character in this material provides crucial evidence for the carrier transport by the shift current mechanism. The profile in the N phase can be explained as the diffusion current arising from the difference in the mobilities between photogenerated electrons and holes known as the Dember effect[43]. Because the amplitude of the photocurrent in the N phase is much smaller than that in the I phase, the contribution of this effect causing the asymmetric photocurrent position profile is negligible in the photovoltaic effect in the I phase. It was very recently reported that the theoretical calculation of the shift current induced by local photoexcitation in a 1D ferroelectric chain based on the Rice–Mele model reproduces our experimental results well[44]. The robust carrier transport without decay for long distances is the most important feature of the shift current.

**Discussion**

Thus far, we have described the fairly large steady-state photocurrent in the I phase. The photocurrent, however, rapidly vanishes when the measurement temperature is decreased from $T_C$ (Fig. 2c). This result appears to be inconsistent with the picture of the shift current mechanism, for which a temperature-independent current is expected once the phase transition occurs because the shift current is determined only by the band structure.

The measurement of the photocurrent response to a pulse light provides insight into the reason for this puzzling temperature

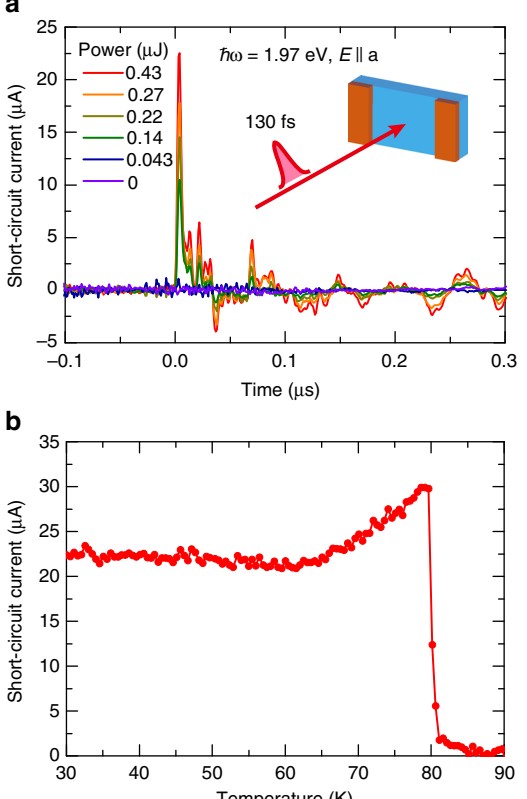

**Fig. 5** Photocurrent response for pulse light. **a** Transient photocurrent response measured at 30 K for varying laser intensity. The photon energy was 1.97 eV, and the light polarization was parallel to the *a*-axis. **b** Temperature dependence of the peak amplitude of the pulse photocurrent for light polarization parallel to the *a*-axis

dependence under CW light irradiation. It is known that the shift current shows an ultra-fast response for pulse light, whose temporal waveform follows that of a pump pulse[45, 46]. Figure 5a presents the transient photocurrent responses for pulse light in the I phase measured using a 130-fs-width pulse laser as a light source. They initially exhibit a pulse current with fast decay and subsequent oscillating components. The first component of the photocurrent can be attributed to the shift current. We confirmed that the magnitude of this component was almost proportional to the laser power. Figure 5b shows the temperature dependence of the peak amplitude of the pulse current. The pulse photocurrent sharply appears at $T_C$, similar to the photocurrent induced by CW light; however, it exhibits a small temperature dependence and remains finite at low temperatures in contrast to the result for CW light. The discrepancy between the pulse and CW lights implies the existence of capacitance components, most likely at the sample–electrode interfaces because of the Schottky barrier formation. The impulsive ac photocurrent can pass through them, whereas the dc photocurrent flow is prevented, particularly at low temperatures. The barrier height estimated using the thermionic emission model is 40 meV. However, the open-circuit photovoltage is not affected by the interfacial resistance but instead by the bulk resistance, which would be the reason for the non-exponential temperature dependence as discussed in Supplementary Note 1.

The results indicate that the energy conversion efficiency in TTF-CA can be greatly improved by tuning the work function of the electrode and by reducing the underlying capacitance components. Thin film growth is another necessary direction to

pursue to improve the efficiency[47]. In addition to applications for energy harvesting, the ultra-fast response time of the shift current is considered useful for many devices including sensors and actuators[48]. This study suggests that strongly correlated electron systems exhibiting electronic ferroelectricity, such as organic CT complexes, show great potential for novel optoelectric devices.

## Methods

**Crystal growth**. Commercially available TTF and CA were purified by repeated recrystallization and sublimation in a vacuum. Single crystals were obtained by cosublimation of the purified TTF and CA in a sealed glass tube.

**Pyroelectric current measurements**. The pyroelectric current along the *a*-axis was measured upon increasing the sample temperature at a rate of 5 K min$^{-1}$ under zero electric field. The measurements were performed after cooling in poling electric fields ($E_{pole}$) of $\pm 2$ kV cm$^{-1}$ and without the poling procedure ($E_{pole} = 0$).

**Photovoltaic measurements**. The photovoltaic properties of TTF-CA were measured by irradiating white light from a solar simulator (HAL-120, Asahi Spectra). The intensity of the light was 0.1 W cm$^{-2}$. The temperature dependences of the short-circuit photocurrent and open-circuit photovoltage were measured in warming processes after the crystal was poled by cooling in a field of $+2$ kV cm$^{-1}$.

**Optical conductivity and photocurrent action spectra**. Polarized optical conductivity spectra were obtained by the Kramers–Kronig analysis of the reflectivity spectra. The reflectivity spectra were measured using a Fourier-transform-type spectrometer in the infrared region and a monochromator-type spectrometer in the visible and ultraviolet regions. The photocurrent action spectra were obtained by measuring the photocurrent under zero electric field by irradiating the sample with monochromized light from a xenon lamp or a halogen–tungsten lamp. The measurements were performed after the poling procedure with a cooling field of 2 kV cm$^{-1}$.

**Photocurrent response for pulse light**. A Ti:sapphire regenerative amplifier system with an optical parametric amplifier operating at 1 kHz was used as the light source for the pulse light measurements. The pulse width was 130 fs, and the spot size was ~1 mmϕ. The polarization of the light was controlled using achromatic wave plates. The short-circuit photocurrent was measured using a pre-amplifier (bandwidth = 200 MHz).

**Data availability**. The authors declare that the data supporting the findings of this study are available within the article and its Supplementary Information.

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

## Acknowledgements

The authors would like to thank S. Ishibashi, N. Nagaosa, H. Ishizuka, J. Fujioka, and M. Sotome for stimulating discussions. This work was partly supported by JSPS KAKENHI (15H05426 and 25220709) and PRESTO JST (JPMJPR16R5).

## Author contributions

M.K. and Y.T. conceived and coordinated the project. S.H. synthesized the single crystals of TTF-CA. M.N. and F.K. performed measurements of the photovoltaic effects. M.N. and T.K. measured the pyroelectric current. M.N. and N.O. measured the transient photocurrent response for pulse light. M.N. wrote the manuscript with input from all the authors.

## Additional information

**Competing interests:** The authors declare no competing financial interests.

