## [Peer Review File · Nature Communications]

Reviewers' comments:

Reviewer #1 (Remarks to the Author):

In this manuscript the authors show convincing evidence for the first observation of a bulk photovoltaic effect (BPVE) in the organic charge transfer compound TTF-CA, which is a ferroelectric below its so-called neutral-ionic transition at 81 K.

The most interesting finding of this work may be that the decisive factor for the very sizable BPVE could be the large dominance of the electronic contribution to the polarization in TTF-CA over its ionic contribution. By demonstrating that the short-circuit current is largely independent of the position of the illuminated area in the ionic phase, the authors present a clear case for the BPVE in a non-centrosymmetric organic charge transfer material. This independence of position has recently been analyzed in theoretical work by Ishizuka and Nagaosa which adds further support to the conclusions of this work.

I have no doubts that the findings presented in this manuscript are relevant and will be interesting to others in the community. Nevertheless, a few points should be addressed before the paper can be accepted for publication:

(1) Language: it is recommended that the authors carefully check their use of articles. In line 65, I would assume that the conjunction "or" should be changed into "and", as the direction of the photocurrent in BaTiO₃ depends on both, polarization and photon energy. Similarly, in line 100 "In reality" is presumably meant to mean "in actuality". Being not a native speaker myself, I am hard-pressed in giving advice on language issues. However, I believe the authors can do better.

(2) It is not clear to me how the open-circuit photo-voltage was measured. Has the crystal been poled by field-cooling, followed by a warm-up in zero field? Please explain in the methods section.

(3) In line 138 the authors write "... of dc photocurrent was kept constant ...". I believe that the correct phrase would be "... of dc photocurrent was constant ...", since otherwise the impression arises that active measures have been taken to keep the photocurrent constant, which is certainly not intended.

(4) It would be good to replace the schematic in Fig. 2a by a photograph of the contacted single crystal. This would have the added value that the reader can judge himself whether electrode misalignment may be the reason for the observed small perpendicular photocurrent (see line 152/153).

(5) In line 169: be more precise in writing "...by charge carrier recombination and phonon scattering..." instead of "...by the recombination and phonon scattering..."

(6) Line 178: Replace "On the contrary" by "In contrast"

(7) Is there any good reason not to shift the Fig. concerning the pumped laser light data into the main paper? The results are discussed there anyway. In this case one could relinquish the supplementary material; which is - to my mind - an advantage for the reader.

(8) Line 202/203: I would recommend to be less enthusiastic about the possible use of BPVE materials, such as TTF-CA, for high-efficiency photovoltaic devices. The observed shift current density of 1.6 micro-A/cm² is quite small as compared to typical values of commercial Si solar cells (28 - 35

mA/cm²). On a second note, with regard to possible use in devices, the thin film morphology would be advantageous. This, at least, has already been shown to work for TTF-CA (see Mater. Res. Expr. 1, 046303 (2014)).

(9) Looking up reference 43 I found that the reference is not correct (paper not found on arXiv). However, the corresponding work has by now already been published in New J. Phys. 19, 033015 (2017); please correct. It might be advisable to carefully check the other references, too.

Reviewer #2 (Remarks to the Author):

In this paper, the authors measured the photo responses of the organic molecular crystal TTF-CA, and found a fairly large short-circuit response under zero-bias. The observed large open-circuit voltage and the position dependence of the photo-current induced by local excitation further illustrate the significant role of the shift current in this material. This work is an important development in the fields of the bulk photovoltaic and the organic ferroelectric solar cells. There are several theoretical work predicting large shift current in low-dimension materials [Ref 17, Ref 43, J. Phys. Chem. C, 121,6500(2017)], and the bulk photovoltaic effect has gained renewed interests in both theory and application. This paper is thus particularly timely and likely to attract a considerable attention from a wider audience. The paper is clear and well-written, and it can be accepted, provided the authors consider the following remarks.

In the discussion part, the authors discussed the effect of the temperature to the short-circuit current. Indeed, the temperature effect is sort of confusing. But it would be useful as a complete paper to include the temperature effect to the open-circuit voltage.

Furthermore, the authors think that the decrease of short-circuit current when reducing the temperature may be because of the crystal-electrode interface, which shows exponential lineshape. If this is the case, should the open-circuit voltage show the similar lineshape, instead of the linear relation to the temperature? Does it mean other contributions besides shift current may contribute to the photocurrent, such as exciton, free carrier or something else?

Minor suggestion:

1. The authors need a brief explanation about the spikes in the optical conductivity below 0.25 eV in Fig. 3a.
2. For Fig. 2c, what is the meaning of "(x-1)" in the upper left of this figure?
3. Sample thickness should be provided.

This may be outside the scope of this paper, but a measured sinuous photocurrent change with respect to the rotation of polarized light will be more convincing to illustrate the role of the shift current in this material as the authors already mentioned in the main text.

REVIEWERS' COMMENTS:

Reviewer #2 (Remarks to the Author):

I am happy with the answers provided by the authors.

The revised manuscript is a greatly enhanced report. I will suggest to accept.

As a final suggestion, the authors could consider putting the answers to my question 1 to the supplementary material to help other readers understand.

Response to Reviewers' comments

Manuscript #: NCOMMS-17-07440

Paper title: Shift current photovoltaic effect in a ferroelectric charge-transfer complex

Authored by: M. Nakamura, S. Horiuchi, F. Kagawa, N. Ogawa, T. Kurumaji, Y. Tokura, and M. Kawasaki

Reviewer #1

All of the authors cordially thank the reviewer for his/her in-depth review and appreciating the value of this paper, and also for the constructive suggestions. In the followings, we provide our point-by-point responses to the reviewer's comments. The responses are reflected in the revised manuscript. A professional English language editing service was employed this time to eliminate possible grammatical errors in English and to improve the readability.

1. Language: it is recommended that the authors carefully check their use of articles. In line 65, I would assume that the conjunction "or" should be changed into "and", as the direction of the photocurrent in BaTiO₃ depends on both, polarization and photon energy. Similarly, in line 100 "In reality" is presumably meant to mean "in actuality". Being not a native speaker myself, I am hard-pressed in giving advice on language issues. However, I believe the authors can do better.

We correct those grammatical errors in English in the revised manuscript. Thank you for pointing them out.

2. It is not clear to me how the open-circuit photo-voltage was measured. Has the crystal been poled by field-cooling, followed by a warm-up in zero field? Please explain in the methods section.

We add a following sentence in the methods section of the revised manuscript: "The temperature dependence of short-circuit photocurrent and open-circuit photovoltage were measured in warming processes after the crystal was poled by cooling in a field of 2 kVcm⁻¹."

3. In line 138 the authors write "... of dc photocurrent was kept constant ...". I believe that the correct phrase would be "... of dc photocurrent was constant ...", since otherwise the impression arises that active measures have been taken to keep the photocurrent constant, which is certainly not intended.

We correct the error in the revised manuscript. Thank you for the remark.

4. It would be good to replace the schematic in Fig. 2a by a photograph of the contacted single crystal. This would have the added value that the reader can judge himself whether electrode misalignment may be the reason for the observed small perpendicular photocurrent (see line 152/153).

Thank you for the suggestion. In the revised manuscript, we add an optical microscope image of the actual sample in Fig. 2a as shown below.

Fig. 2a

5. In line 169: be more precise in writing "...by charge carrier recombination and phonon scattering..." instead of "...by the recombination and phonon scattering..."

We correct that part according to the reviewer's suggestion. Thank you for the suggestion.

6. Line 178: Replace "On the contrary" by "In contrast"

We correct the error in the revised manuscript. Thank you for pointing it out.

7. Is there any good reason not to shift the Fig. concerning the pumped laser light data into the main paper? The results are discussed there anyway. In this case one could relinquish the supplementary material; which is - to my mind - an advantage for the reader.

Following the reviewer's suggestion, we transfer the Figure in the supplementary material to the main text as Fig. 5. Thank you for the constructive suggestion.

8. Line 202/203: I would recommend to be less enthusiastic about the possible use of BPVE materials, such as TTF-CA, for high-efficiency photovoltaic devices. The observed shift current density of 1.6 micro-A/cm² is quite small as compared to typical values of commercial Si solar cells (28 - 35 mA/cm²). On a second note, with regard to possible use in devices, the thin film morphology would be advantageous. This, at least, has already been shown to work for TTF-CA (see Mater. Res. Expr. 1, 046303 (2014)).

We thank the reviewer for the kind advices. As the reviewer points out, the observed photocurrent in the bulk crystal of TTF-CA is still quite small compared to that in commercial solar cells. In the revised manuscript, we rephrase the sentence from "The robust carrier transport without decay for long distance would be the most important advantage of the shift current toward the realization of high-efficiency photovoltaic devices." to "The robust carrier transport without decaying for long distance is the most important feature of the shift current."

In addition, we add a following sentence in the discussions section to mention the importance of high-quality film for possible use in devices with citing the paper Mater. Res Expr. 1, 046303 (2014): "Thin film growth is another necessary direction to pursue to improve the efficiency⁴⁷."

9. Looking up reference 43 I found that the reference is not correct (paper not not be found on arXiv). However, the corresponding work has by now already been published in New J. Phys. 19, 033015 (2017); please correct. It might be advisable to carefully check the other references, too.

We thank the reviewer for pointing out the error in the reference. We corrected the reference, and also carefully checked all the other references.

Reviewer #2

All of the authors cordially thank the reviewer for his/her in-depth review and giving us constructive comments and suggestions. In the followings, we provide our responses to all of the reviewer's comments one-by-one.

In this paper, the authors measured the photo responses of the organic molecular crystal TTF-CA, and found a fairly large short-circuit response under zero-bias. The observed large open-circuit voltage and the position dependence of the photo-current induced by local excitation further illustrate the significant role of the shift current in this material. This work is an important development in the fields of the bulk photovoltaic and the organic ferroelectric solar cells. There are several theoretical work predicting large shift current in low-dimension materials [Ref 17, Ref 43, J. Phys. Chem. C, 121,6500(2017)], and the bulk photovoltaic effect has gained renewed interests in both theory and application. This paper is thus particularly timely and likely to attract a considerable attention from a wider audience. The paper is clear and well-written, and it can be accepted, provided the authors consider the following remarks.

We appreciate that the reviewer highly appreciates the value of our paper. In the revised manuscript, we add a recent paper on a theoretical calculation of shift current in organic polymers (J. Phys. Chem. C, **121**, 6500 (2017)) to the references.

In the discussion part, the authors discussed the effect of the temperature to the short-circuit current. Indeed, the temperature effect is sort of confusing. But it would be useful as a complete paper to include the temperature effect to the open-circuit voltage.

Furthermore, the authors think that the decrease of short-circuit current when reducing the temperature may be because of the crystal-electrode interface, which shows exponential lineshape. If this is the case, should the open-circuit voltage show the similar lineshape, instead of the linear relation to the temperature? Does it mean other contributions besides shift current may contribute to the photocurrent, such as exciton, free carrier or something else?

Thank you for the comment. Although we have not completely understood the temperature dependence of short-circuit photocurrent and open-circuit photovoltage, here we first discuss the former on the basis of the data analysis and then discuss the latter with assuming an equivalent circuit. Since they are still too preliminary to expose to the readers, we just add a following sentence in the revised manuscript: "However, the open-circuit photovoltage is not affected by

the interfacial resistance but instead by the bulk resistance, which would be the reason for the non-exponential temperature dependence.”

Figure R1 shows the data of short-circuit photocurrent as a function of temperature on a semi-logarithmic scale and their fitting to a thermionic emission model assuming the contact involves a Schottky barrier. The data are well fitted to the formula of

$$J = AT^2 \exp\left(-\frac{e\Phi_B}{kT}\right) \quad (1),$$

where e is the elementary charge, Φ_B the barrier height, k the Boltzmann constant, T temperature, and A the Richardson constant, resulting in $\Phi_B \sim 40$ meV. Considering its narrow bandgap of about 500 meV, the barrier height is within reasonable range. Therefore it is most likely that the short-circuit current is limited by the contact resistance due to a Schottky barrier.

Fig. R1: Temperature dependence of short-circuit photocurrent on a semi-logarithmic scale. The black line is a result of the fitting by the thermionic emission model.

Having established a quasi-exponential temperature dependence of contact resistance, we now discuss the temperature dependence of the open-circuit photovoltage with assuming an equivalent circuit model. Figure R2 displays the simplest equivalent circuit that may represent device condition during photovoltaic effect measurements, where I_{shift} is shift current which works as a current source, I_{obs} is an observed current, V is a voltage between two electrodes, and R_{bulk}^* and R_{contact}^* are bulk and contact resistances for photocurrent under illumination, respectively. These parameters are related by the following equation,

$$I_{\text{obs}} = \frac{1}{R_{\text{bulk}}^* + R_{\text{contact}}^*} V + \frac{R_{\text{bulk}}^*}{R_{\text{bulk}}^* + R_{\text{contact}}^*} I_{\text{shift}} \quad (2).$$

The open-circuit photovoltage (short-circuit photocurrent) is output voltage (current) when $I_{\text{obs}} = 0$ ($V = 0$). Then, V_{OC} and I_{SC} are given by

$$V_{OC} = -R_{\text{bulk}}^* I_{\text{shift}} \quad (3),$$

$$I_{SC} = \frac{R_{\text{bulk}}^*}{R_{\text{bulk}}^* + R_{\text{contact}}^*} I_{\text{shift}} \quad (4).$$

Fig. R2: An equivalent circuit for the sample condition during measurements of photovoltaic effect

Suppose I_{shift} is temperature independent and $R_{\text{bulk}}^* \ll R_{\text{contact}}^*$, Eq. 4 implies that I_{SC} exponentially decays in low temperature as was shown in Fig. R1, whereas Eq. 3 implies that V_{OC} is proportional to R_{bulk}^* which can have non-exponential temperature dependence. We tried a measurement of R_{bulk}^* , but the larger R_{contact}^* prevents the accurate evaluation of R_{bulk}^* . We will need non-contact techniques such as a terahertz spectroscopy to derive accurate value of R_{bulk}^* .

As pointed out by the reviewer, we cannot rule out other possibilities, such as exciton, free carriers, and defects. These can be serious causes of the photocurrent suppression. However, our results exhibit clear difference between the temperature dependence of photocurrent for CW light and that for pulse light, indicating that the most dominant origin is the interfacial potential barrier. To completely understand the mechanism of the temperature dependences, further theoretical advances will be necessary enabling the calculation of shift current for more realistic conditions of materials including free carriers and defects.

Response to minor suggestions

1. *The authors need a brief explanation about the spikes in the optical conductivity below 0.25 eV in Fig. 3a.*

The several peaks observed in the optical conductivity spectra below 0.25 eV are related to intramolecular vibration modes of TTF and CA molecules (Please see M. Dressel *et al.* Crystals **7**, 17 (2017)). To avoid possible readers' confusion, we add a following sentence in the Figure caption for Fig. 3a: "The several peaks observed in low photon-energy region are related to intramolecular vibration modes of TTF and CA molecules."

2. For Fig. 2c, what is the meaning of "(x-1)" in the upper left of this figure?

The meaning of ($\times -1$) is that the sign of photocurrent is negative. The temperature dependence of the short-circuit photocurrent was measured in warming processes after the sample was poled by cooling under a positive electric field. Then, negative short-circuit photocurrent appears (please see Fig. 2b). To avoid possible readers' confusion, we modify the vertical axis in Fig. 2c in the revised manuscript as shown below.

Fig. 2c

3. Sample thickness should be provided.

Sample thickness is 0.2 mm. In the revised manuscript, we denote the sample thickness in the schematic of the sample structure shown in Fig. 2a.

4. This may be outside the scope of this paper, but a measured sinuous photocurrent change with respect to the rotation of polarized light will be more convincing to illustrate the role of the shift current in this material as the authors already mentioned in the main text.

Thank you for the suggestion. The photocurrent spectra for light polarizations parallel and perpendicular to a -axis are shown in Fig. 3b. They exhibit distinct light polarization dependence. It is readily expected that photocurrent shows sinusoidal dependence between the two polarization directions. Although it will be necessary to examine more detailed polarization dependence in the future to provide further convincing evidence for anisotropic characteristics of the shift current, we consider that Fig. 3b is a clear hallmark of the anisotropic photocurrent and sufficient for the present paper.

Summary of changes in the revised manuscript

1. Errors in English words pointed out by the Reviewer #1 are corrected.
2. In the explanation of photovoltaic measurement in Methods part, we add a following sentence: “The temperature dependence of short-circuit photocurrent and open-circuit photovoltage were measured in warming processes after the crystal was poled by cooling in a field of 2 kVcm^{-1} .”
3. In Fig. 2a, we add a polarized optical microscope image of the actual sample.
4. The figure originally shown in the supplementary material is transferred to the main text as Fig. 5. We also add a description on the measurement of photocurrent response for pulse light in the Methods part.
5. The description on the possible use of high conversion efficiency photovoltaic devices on line 202-203 of the original manuscript is rephrased as follows: “The robust carrier transport without decay for long distances is the most important feature of the shift current.” In addition, a following sentence is added in the discussions section of the revised manuscript: “Thin film growth is another necessary direction to pursue to improve the efficiency⁴⁷.”
6. References 18 and 47 are added.
7. In discussions section, we add a following sentence: “However, the open-circuit photovoltage is not affected by the interfacial resistance but instead by the bulk resistance, which would be the reason for the non-exponential temperature dependence.”
8. A following sentence is added in the caption for Fig. 3a: “The several peaks observed in low photon-energy region are related to intramolecular vibration modes of TTF and CA molecules.”
9. Figure 2c is slightly modified.
10. In the last paragraph of introduction section, we add following sentences to satisfy the format requirements of Nature Communications: “We have examined the photovoltaic properties of TTF-CA and observed that sizable zero-bias photocurrent appears with visible-light irradiation in the I phase. We have also revealed an anomalously long travel

distance of photocarriers. These results provide clear indications of the shift current first observed in organic CT complexes.”